# 3D Spongin Scaffolds as Templates for Electro-Assisted Deposition of Selected Iron Oxides

**DOI:** 10.3390/biomimetics9070387

**Published:** 2024-06-25

**Authors:** Krzysztof Nowacki, Anita Kubiak, Marek Nowicki, Dmitry Tsurkan, Hermann Ehrlich, Teofil Jesionowski

**Affiliations:** 1Institute of Chemistry and Technical Electrochemistry, Poznan University of Technology, Berdychowo 4, 60-965 Poznan, Poland; 2Faculty of Chemistry, Adam Mickiewicz University, Uniwersytetu Poznanskiego 8, 61-614 Poznan, Poland; anikub@amu.edu.pl; 3Center of Advanced Technology, Adam Mickiewicz University, Uniwersytetu Poznanskiego 10, 61-614 Poznan, Poland; marek.nowicki@amu.edu.pl (M.N.); herehr@amu.edu.pl (H.E.); 4Faculty of Materials Engineering and Technical Physics, Poznan University of Technology, Piotrowo 3, 60-965 Poznan, Poland; 5Institute of Nanoscale and Biobased Materials, TU Bergakademie Freiberg, Gustav-Zeuner-Str. 3, 05999 Freiberg, Germany; 6Institute of Chemical Technology and Engineering, Poznan University of Technology, Berdychowo 4, 60-965 Poznan, Poland

**Keywords:** spongin, iron oxides, electrodeposition, biocorrosion, biomimetics, *Hippospongia communis*

## Abstract

The skeletons of marine sponges are ancient biocomposite structures in which mineral phases are formed on 3D organic matrices. In addition to calcium- and silicate-containing biominerals, iron ions play an active role in skeleton formation in some species of bath sponges in the marine environment, which is a result of the biocorrosion of the metal structures on which these sponges settle. The interaction between iron ions and biopolymer spongin has motivated the development of selected extreme biomimetics approaches with the aim of creating new functional composites to use in environmental remediation and as adsorbents for heavy metals. In this study, for the first time, microporous 3D spongin scaffolds isolated from the cultivated marine bath sponge *Hippospongia communis* were used for electro-assisted deposition of iron oxides such as goethite [α-FeO(OH)] and lepidocrocite [γ-FeO(OH)]. The obtained iron oxide phases were characterized with the use of scanning electron microscopy, FTIR, and X-ray diffraction. In addition, mechanisms of electro-assisted deposition of iron oxides on the surface of spongin, as a sustainable biomaterial, are proposed and discussed.

## 1. Introduction

Three-dimensional centimeter-scale structures with microporous ornamentation, both synthetic and biological in origin, are the basis of modern scaffolding strategies [1,2] for further applications in biomedicine and technologies. Due to the global pollution of the world’s oceans with nano- and microplastics, the search for alternative raw materials for the production of such 3D matrices has recently intensified. Classic examples of such structures of natural origins include both plant (fibrous *Luffa* sponge) [3,4,5,6] and diverse marine sponges (bath sponges) [7,8,9,10], the skeletons of which are ready-to-use matrices. These natural scaffolds are well recognized renewable sources of pre-structured biological materials, but they differ in their chemistry. In contrast to the cellulose-based *Luffa* sponge, the skeletal constructs of bath sponges are made of the biopolymer spongin. This mechanically stable and elastic biomaterial contains collagen-like as well as keratin-like domains [11]. Due to the presence of iodine, chlorine, and bromine in concentrations that range between 1% and 6%, this structural biocomposite is still chemically non-identified [9].

The high resistance of spongin to various enzymes, acids, and high temperatures has determined the success of this biomaterial in modern practical materials science, including extreme biomimetics [12,13,14,15]. Purified spongin scaffolds have been used to produce composite materials under hydrothermal synthesis conditions [13,14,16], such as catalysts [14] being used as drug carriers [17], and as unique substrates for the production of 3D graphite [18]. Additionally, spongin matrices that were originally covered with iron oxides due to naturally occurring biocorrosion in sponges inhabiting diverse iron-based artificial constructs (such as shipwrecks) have come to the attention of researchers [19,20,21]. On the one hand, these kinds of biocomposite materials can be used in practice in a finished form; on the other hand, they serve as an example motivating the creation of new iron-containing 3D functional materials in the form of microporous sponges.

Recently, two new studies on combining spongin with iron have been reported. The first [22] addressed the complexity of marine spongin chemistry, specifically focusing on the interaction of this biopolymer with iron ions, leading to the development of a new composite material named Iron–Spongin. This biomimetic approach not only elucidates the mechanism of lepidocrocite [γ-FeO(OH)] formation but also proposes a potential use of the resulting composite as an electrochemical dopamine sensor. In the second study [12], a spongin from the marine bath sponge *Hippospongia communis* was used as a template to create a microporous composite containing goethite [α-FeO(OH)]. This innovative technique employed an extreme biomimetic approach, utilizing iron powder, crystalline iodine, and fibrous spongin for the first time. The resulting composite offers promising applications in biomedicine and bio-inspired materials science, again including the electrochemical detection of dopamine.

Electrodeposition is a versatile and widely employed technique for the controlled deposition of materials on conductive substrates through the application of an electrical current. The method is particularly suited to materials science applications, such as the coating of surfaces with thin layers of metals or metal compounds [23,24,25,26]. By adjusting parameters such as the voltage, current density, and electrolyte solution composition, researchers can precisely control the thickness, composition, and morphology of the deposited layers. Previously, electrodeposition has been employed with a range of metals, including iron compounds [27,28,29], to coat porous materials [30,31,32]. Such coatings are of great importance in improving the mechanical strength, corrosion resistance, and functional properties of the base materials. One of the key advantages of electrodeposition is its ability to ensure the uniform coating of complex geometric surfaces, providing comprehensive and homogeneous coverage even in porous substrates. Electrodeposition of metals such as copper on 3D chitin-based demosponge scaffolds using an extreme biomimetics approach has already been reported [33]. It is known that spongin in its unmodified form does not conduct electrical currents. Therefore, despite the technique’s many advantages, the possibility of using electrodeposition to create spongin–metal complexes has previously been regarded as questionable. However, inspired by previous works [12], we have decided to use electrolysis as a convenient method to arrange an oxidative/reductive environment and thus accelerate the electroless deposition of iron compounds on the surface of spongin scaffolds. 

In this study, for the first time, we used 3D spongin scaffolds isolated from the cultivated marine bath sponge *H. communis* for the electro-assisted deposition of selected iron oxides.

## 2. Materials and Methods

### 2.1. Materials

Purified, mineral-free spongin scaffolds from the marine demosponge *Hippospongia communis* (Lamarck, 1814) were acquired from INTIB GmbH (Freiberg, Germany). Sodium sulfate (Na_2_SO_4_, ≥99.7%), supplied by Sigma-Aldrich (Burlington, MA, USA), was used for the preparation of the aqueous electrolyte solution. Redistilled water was used to prepare all aqueous solutions. Goethite standard was obtained from Sigma-Aldrich (Burlington, MA, USA) and lepidocrocite standard from Nanochemazone (Leduc, AB, Canada). 

### 2.2. Electrolyzer Setup

A view of the experimental system and the design scheme of the electrolyzer for the deposition of iron oxide on spongin scaffolds is shown in Figure 1A,B.

The CEM (cation exchange membrane) electrolyzer (Figure 1A) consisted of two cylindrical poly(propylene) chambers (50 mL each) separated by a PET-reinforced cation exchange membrane (Fumasep^®^ FKS-PET-130, Fumatech BWT GmbH, Bietigheim-Bissingen, Germany). Both electrodes were made of steel wire (ø 0.5 mm) and were 9 cm in length, with a distance between them of about 10 cm. They were connected with a DC power supply (Aim-TTi PLH120, Aim-TTi, Huntingdon, UK). An aqueous sodium sulfate solution (1.5 M) with an initial temperature of 30 °C was utilized as both the anolyte and catholyte, and an Elmetron CPC-411 digital pH meter (Elmetron, Zabrze, Poland) was used to control the pH of the electrolyte.

### 2.3. Deposition of Iron Oxide

In pretreatment, the *H. communis* spongin scaffold (Figure 2A) was cut into 1.0 g pieces (Figure 2B,C) and rinsed repeatedly with redistilled water to eliminate major solid impurities and water-soluble salts of marine origin.

The electro-assisted deposition process consisted of two steps (the initial setup of the electrolyzer is shown in Figure 1B).

Step 1—Pretreatment (Figure 3A). First, to ensure acidic/alkaline conditions within the electrolyzer chambers and to dissolve iron through an oxidation reaction on the anode, electrolysis of 1.5 Na_2_SO_4_ was performed for 1 h (12 V, 0.5 A). This resulted in a solution denoted as anolyte I, with pH = 1.5 (50 °C) and a rust color and a colorless catholyte I with pH = 12.0 (50 °C). 

Step 2—Deposition (Figure 3B). In order to cover the sponge samples with iron compounds, both specimens were placed in the electrolyzer chambers. Sample S1 was soaked in anolyte I and sample S2 in catholyte I. The polarization of the electrodes was then changed, and electrolysis was carried out for 1 h (16 V, 0.5 A). 

Due to the polarity change within the chamber with sample S1, the expected reaction was an exchange between the iron (III) sulfate (VI) formed in the pretreatment step and hydroxide ions formed in the cathodic reduction of the water molecule, resulting in iron (III) hydroxide (Figure 4A). Simultaneously, in the chamber with sample S2, the alkaline conditions and electro-oxidation of the steel anode were expected to result in the formation and deposition of iron (III) oxide–hydroxide on the spongin (Figure 4B).

### 2.4. Characterization Techniques

#### 2.4.1. Digital Microscopy

The acquired samples were examined and assessed employing a high-definition imaging setup, specifically the VHX-6000 digital optical microscope (Keyence, Osaka, Japan), outfitted with a VH-Z20R zoom lens allowing for magnifications up to 200 times. Additionally, the investigation utilized a VHX-7000 digital optical microscope (Keyence, Japan), equipped with VHX-E20 and VHX-E100 zoom lenses, providing magnification capabilities of up to 100 times and 500 times, respectively.

#### 2.4.2. Scanning Electron Microscopy (SEM) with Energy-Dispersive X-ray Analysis (EDX)

The chemical composition and topography of the spongin scaffold samples were determined through SEM-EDX analyses. These assessments were conducted using a low-vacuum scanning electron microscope, specifically the JEOL JSM-6610LV model with a LaB6 cathode. This device was additionally fitted with an energy-dispersive X-ray spectrometer comprising a 10 mm^2^ silicon drift detector (SDD), the X-Flash 6|10, from Bruker, Co. (Berlin, Germany).

#### 2.4.3. Fourier Transform Infrared Spectroscopy (FTIR)

The FTIR spectra for the materials under study were captured with a Nicolet iS50 spectrometer (Thermo Fisher Scientific Co., Hillsboro, OR, USA). Each spectral measurement utilized an integrated attenuated total reflectance (ATR) feature. The examinations spanned a wavelength ranging from 4000 to 400 cm^−1^.

#### 2.4.4. X-ray Diffraction (XRD)

X-ray analysis of the materials was performed using a SmartLab Rigaku powder diffractometer (Tokyo, Japan), equipped with a CuKα source. The examination covered a 2θ range from 3 to 80, with a scan step of 0.01 and a scan speed of 4 degrees per minute.

## 3. Results

### 3.1. Morphology and Microstructure of Processed Spongin Scaffolds

The digital microscopy images (Figure 5) reveal a significant difference in morphological characteristics between the original spongin fibers and those of the treated samples. 

The original spongin sample displays a yellowish, reticulated microfibrous network. In contrast, samples S1 (Figure 5B) and S2 (Figure 5D) exhibit a uniform rusty hue, which is consistent with the deposition of iron oxide closely associated with the organic matrix of the spongin fibers. The iron oxide deposition is visible as a cohesive surface layer, indicating a significant interaction between the oxide and the spongin components. Additionally, upon closer inspection of sample S1 (Figure 5C), a fiber with a fine, pale-yellow deposit is visible, suggesting a distinct oxide incorporation phase that survived even after sonication at room temperature for 1 h. The rust-colored deposit is most noticeable in sample S2, where the fibers are tightly compacted and show a texture that suggests greater oxide accumulation.

The SEM images in Figure 6 provide a visual presentation of the morphology of the spongin fibers before and after the electro-supported deposition of iron-based compounds. Figure 6A,B show the natural, unmodified surface of the pristine spongin fibers. The fibers are seamlessly intertwined, displaying consistent texture and integrity, as evidenced by their smooth surfaces under high magnification.

In contrast, Figure 6C,D illustrate the altered appearance of sample S1. Figure 6C shows a rough topography with crystalline iron oxide deposits that are visible as micron-scale features on the fibers, indicating a localized concentration of iron oxide formation. The magnified view in Figure 6D provides further explanation of this phenomenon, with a distinct crystalline formation being clearly visible at the central point of iron oxide accretion. The crystal structure suggests that there is heterogeneous nucleation, which indicates areas of preferential growth along the fiber surface. 

Figure 6E,F show that sample S2 undergoes a different interaction with iron oxide than sample S1. Instead of the crystalline efflorescence observed in S1, sample S2 has a uniform coating of iron oxide that engulfs the fibers in a thin, granular layer. This uniform coating suggests a more homogeneous interaction with the iron oxide. The high magnification of Figure 6F does not reveal single crystals. Instead, it shows a densely packed layer of iron oxide particles, implying a thorough and uniform modification of the fiber surface.

### 3.2. Chemical Composition of Samples 

The analysis of sample S1 (Figure 7B) reveals a high iron content (60.13 wt%), indicative of significant iron oxide deposition on the substrate. The associated high oxygen peak (29.64 wt%) further substantiates the presence of iron oxide. This suggests the formation of a robust iron oxide layer on the sample surface. In contrast, the EDX analysis of sample S2 (Figure 7C) demonstrates a lower iron content (47.67 wt%). The reduced iron content indicates a less extensive iron oxide deposition, potentially influenced by the alkaline conditions that are present in the catalyst chamber during sample preparation, which may have resulted in the formation of a more dispersed iron oxide layer. The oxygen content of 33.99 wt% in sample S2 confirms the presence of oxides, albeit in a different distribution compared to sample S1. 

A comparison between the untreated spongin and the treated samples (S1 and S2) demonstrates the successful deposition of iron oxide onto the spongin matrix. The untreated spongin exhibits no iron content, while both treated samples contain significant amounts of iron, thereby confirming the efficacy of the deposition process.

The FTIR spectral analysis enables a detailed comparison between the pure spongin and electrolytically treated S1 and S2 spongin samples (Figure 8). The spectrum of untreated spongin contains internal absorption bands which, according to the literature, correspond to the functional groups of spongin [34]. Table 1 lists the newly evident specific bands in the spectral data, including their wavenumbers and assignments.

The treatment process led to the appearance of new bands in the spectra for samples S1 and S2, including a prominent O–H stretching vibration band at 3142 cm^−1^, which is characteristic of oxyhydroxides [35]. The band observed at 570 cm^−1^ is representative of the Fe–O stretching vibrations that are commonly associated with iron oxides [36,37]. The bands at 892 cm^−1^ and 794 cm^−1^ obtained for both samples indicate the presence of goethite (α-FeOOH), as they result from in-plane deflection of surface OH in Fe–OH–Fe [38,39]. The more prominent expression of these bands in sample S1 suggests a more crystalline goethite content than in S2. Additionally, the FTIR spectrum of sample S1 has a pronounced absorption band at 1021 cm^−1^, signifying the presence of lepidocrocite (γ-FeOOH) and suggesting its substantial integration [40]. The absorption features at 1150 cm^−1^ and 740 cm^−1^ are associated with the deformation and bending of OH within γ-FeOOH [41,42]. The marked intensity of the bands at 1021 cm^−1^ and 740 cm^−1^ may indicate the significant presence of a well-crystallized lepidocrocite phase.

Both the S1 and S2 spectra confirm the formation of goethite and lepidocrocite, but S1 exhibits higher crystallinity of these iron oxides, as evidenced by more distinct and intense spectral signatures. These signatures suggest that iron oxides have been successfully implanted into the spongin structure, which may enhance its usefulness in a range of applications.

### 3.3. Characterization of the Crystalline Structure of the Iron Oxide–Hydroxide–Spongin Composites

X-ray diffraction analysis was performed on untreated spongin and the iron oxide-treated samples S1 and S2, and a comparison was made with lepidocrocite and goethite standards. The results provide a clear visualization of the crystalline phases induced by the electrolysis process. The spongin baseline (Figure 9A) shows a largely featureless curve with a broad hump and no sharp peaks, indicating an amorphous structure that is typical of organic materials such as a spongin matrix of marine sponge origin [14,34]. However, small detectable peaks at approximately 21°, 33°, and 36° are present. These peaks might indicate trace amounts of iron oxides [43]. This confirms that, as reported in the literature [19], small amounts of lepidocrocite and goethite can naturally be found in marine sponges. 

Sample S1 (Figure 9B) displays sharp, well-defined peaks that correspond to crystalline iron oxide phases. Specifically, peaks are visible at 2θ values that correspond to the standard positions of lepidocrocite (L) and goethite (G). These include peaks for goethite at approximately 21°, 33°, 34°, 36°, and 61° [43,44,45], as well as peaks for lepidocrocite at around 14°, 27°, and 53° [46,47]. The peaks’ alignment with the lepidocrocite (Figure 9D) and goethite (Figure 9E) standards indicates the presence of these iron oxide hydrates in the sample.

Sample S2 (Figure 9C) also produces peaks at the same 2θ values, indicating lepidocrocite and goethite. However, the peaks’ intensity is significantly lower than in S1, suggesting a lower crystallinity or concentration of these phases in S2. The reduced intensity observed in the case of this sample may be due to differences in deposition conditions, which potentially affected the growth and formation of crystalline iron oxide hydrates. A direct comparison of the peak position and intensity between the treated spongin samples and the standard references clearly shows the successful incorporation of iron oxide hydrates into the spongin fibers during the electrolytic treatment process.

## 4. Discussion

Our previous studies have proven that it is possible to mimic a natural process of biocorrosion occurring in marine sponges habituating near and on iron-based artificial constructs [12,22]. Forced mineralization was employed to create a spongin–goethite composite, using the marine demosponge *H. communis* as a biological 3D blueprint for the crystallization process. The key idea of this approach was to utilize crystalline iodine as an oxidant of metallic iron to produce ferrous ions (Fe^2+^), which were further oxidized by atmospheric oxygen to ferric ions (Fe^3+^) and bound into goethite in a series of crystalline transformations, possibly catalyzed by spongin fibers [12]. The main drawback of this approach was the long treatment time resulting from the oxidation reaction of ferrous to ferric ions under atmospheric oxygen. Therefore, in our current research, we sought an alternative synthesis method for composites of iron-containing minerals and spongin. 

Simultaneously, understanding the principles of biomineralization and implementing technical solutions inspired by this process in modern materials science are among the main current objectives of biomimetics [48,49]. The biocorrosion of steel objects is a widespread phenomenon, identified especially in studies on marine sessile invertebrates [50,51,52]. These organisms have the ability to store inorganic ferric/ferrous ions in the form of mineralized skeletons for use in their survival strategies under appropriate environmental conditions [19,20,21]. Skeletal biopolymers substantially contribute to the formation of iron-based biominerals in various forms, including magnetite (Fe_3_O_4_), hematite (Fe_2_O_3_), goethite [α-FeO(OH)], and lepidocrocite [γ-FeO(OH)]. 

When considering forced redox reactions, electrolysis is a technique that comes immediately to mind. Since spongin in its unmodified form is not an electrical conductor, electroplating with metallic iron or its composites was not previously considered possible [30,31,32]. However, inspired by other studies [53,54], we decided to use a cation exchange membrane electrolyzer as a convenient tool to arrange an oxidative/reductive environment and thus accelerate the electroless deposition of iron minerals on the spongin fibers. The fundamental electrochemical reactions that occur during the electrolysis of a Na_2_SO_4_ aqueous solution (decomposition of water particles) take place on the electrode surfaces, namely on the anode surface,
2H_2_O_(l)_ → O_2(g)_ + 4H^+^_(aq)_ + 4e^−^(1)
and on the cathode surface,
4H_2_O_(l)_ + 4e^−^ → 2H_2(g)_ + 4OH^−^_(aq)_(2)

The result of this redox reaction is an excess of H^+^ ions in the anolyte and an excess of OH^−^ ions in the catholyte [55,56,57]. This method allowed us to implement two strategies.

The first approach (the acidic reduction method) was focused on creating an environment that was rich in Fe^3+^ ions but highly acidic (anolyte I). Briefly, the Fe^2+^ ions from the anode dissolution undergo rapid oxidation by highly concentrated oxygen particles created in the parallel water oxidation reaction in anolyte I [55,56,57]. Therefore, when the spongin sample (S1) is placed in such a solution, its functional groups of amino-acidic origin [19] occur in their protonated forms, and the spongin scaffold itself is saturated with Fe^3+^ ions. Simultaneously, a change in electrode polarity caused a slow increase in pH (due to the cathodic reduction of water particles), allowing for the creation of iron oxide–hydroxides and their binding—possibly through hydrogen bonds—with the functional groups of spongin [57,58].

The second approach (the alkaline oxidation method) was focused on recreating and boosting the alkaline conditions of seawater (catholyte I) by reducing the water particles. Keeping in mind that the amino acids of spongin occur naturally in their anionic form (seawater, pH = 8.1), the formation of spongin–iron composites that are similar to those observed in nature was expected. After establishing alkaline conditions and placing the spongin sample (S2) in catholyte I, the polarity of the electrode was changed, and through alkaline electro-oxidation of the steel anode and electrolysis of water particles, ferrous ions were introduced to the solution (through a similar mechanism as in the case of anolyte I).

The subsequent analysis of the goethite/lepidocrocite–spongin composites gave clear results, confirming that the iron (III) oxide–hydroxides were successfully implanted into the spongin structure in both cases. However, it is notable that there are some significant differences in the samples’ morphology and crystallinity. The acidic reduction method resulted in more crystalline but heterogeneous deposits of iron (III) oxide–hydroxides in the form of efflorescences, which may be caused by the high initial concentration of Fe^3+^ ions followed by their excellent distribution within a spongin sample saturated in anolyte I. Such starting conditions, following the change in the electrode’s polarity, affected the heterogeneous nucleation of the goethite/lepidocrocite, leading to the formation of distinct crystal structures. By contrast, the alkaline oxidation method led to uniform layers of coating with iron oxides and hydroxide. This phenomenon can be explained by the lack of Fe^3+^ ions in the initial solution of highly alkaline catholyte I. The ferric ions were gradually introduced in the second step of this method (after the change in electrode polarity). Therefore, the process of crystallization on the spongin fibers was limited by the diffusion of the iron oxide particles inside the sponge structure. Simultaneously, the alkaline conditions caused the majority of spongin functional groups to take their anionic forms, which allowed them to uniformly coat the fibers with a thin, granular layer of iron oxide–hydroxides. It is also worth noting that compared with the iodine oxidation method (72 h) or the artificial seawater method (30 days) [12,22], the electro-assisted deposition process allows for a significant shortening of the time needed to form composites of spongin and iron (III) oxide–hydroxides, possibly due to the elimination of Fe^2+^ ion oxidation with atmospheric oxygen and its replacement with the much faster oxidation achieved using oxygen produced in situ by the electrolysis of water [55].

Moreover, alongside the formation of goethite–spongin, the electro-assisted synthesis of lepidocrocite was confirmed for the first time. Lepidocrocite is considered less thermodynamically stable than goethite, and its formation is favored by lower temperatures (≤40 °C) and low pH [53]. Therefore, in future research on such materials, the electro-assisted deposition process (especially the acidic reduction method) should be considered as one of the first choices. However, future detailed investigations of the mechanisms occurring during the electro-assisted deposition of goethite/lepidocrocite on spongin fibers are required, since slight changes in the electrolysis parameters (such as potential, current, time, and pH) may lead to increased selectivity of the process and help to form more desirable morphological structures [46,53]. 

## 5. Conclusions

This study successfully demonstrates the feasibility of utilizing 3D spongin scaffolds derived from the cultivated marine bath sponge *H. communis* as templates for the electro-assisted deposition of selected iron oxides, specifically goethite (α-FeO(OH)) and lepidocrocite (γ-FeO(OH)). Detailed characterization using digital microscopy, SEM-EDX, FTIR, and XRD confirmed the effective incorporation of these iron oxides into the spongin structure. The acidic reduction method resulted in more crystalline but heterogeneous deposits, while the alkaline oxidation method produced a uniform granular coating of iron oxides.

The electro-assisted deposition method significantly reduces the time needed to form spongin–iron oxide composites compared with traditional techniques. Our results provide new insights into biomineralization and present a versatile and efficient approach to developing new biomimetic materials with potential applications in biomedicine and bio-inspired materials science. Future work will focus on optimizing the parameters of electro-assisted deposition and exploring the use of other metal oxides to expand the applications of this biomimetic technique.

## Figures and Tables

**Figure 1 biomimetics-09-00387-f001:**
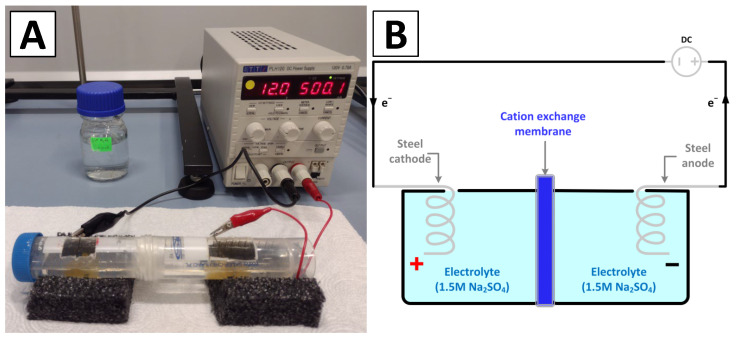
The experimental setup used in this study (**A**) and the general design of the CEM electrolyzer with a Na_2_SO_4_ aqueous solution as the electrolyte (**B**).

**Figure 2 biomimetics-09-00387-f002:**
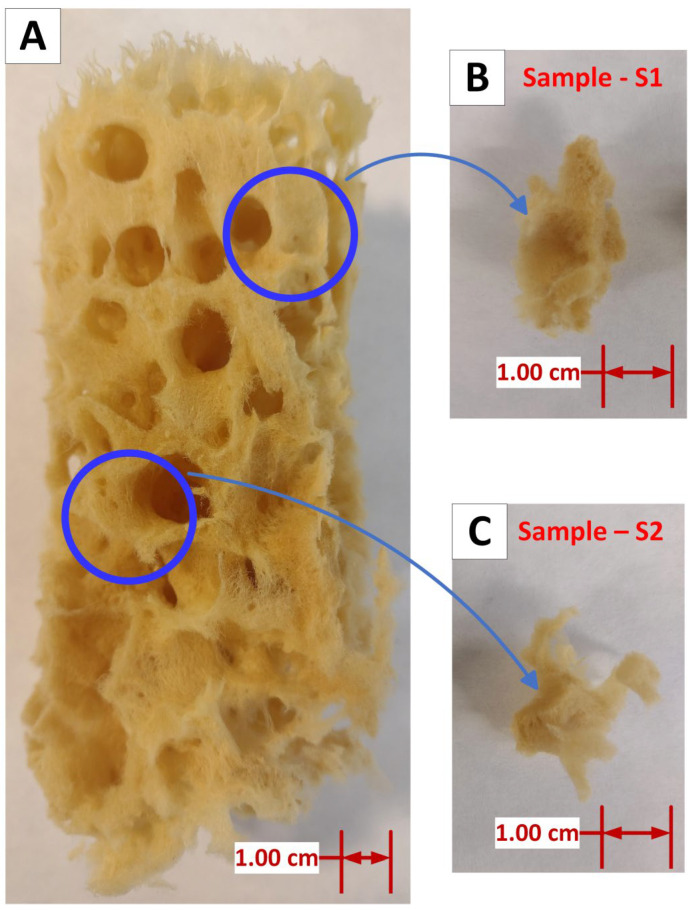
Overview of the 3D *H. communis* spongin scaffolds used in this study (**A**) and a close-up view of the two samples used in electrolysis as the cathode chamber batch (**B**) and the anode chamber batch (**C**).

**Figure 3 biomimetics-09-00387-f003:**
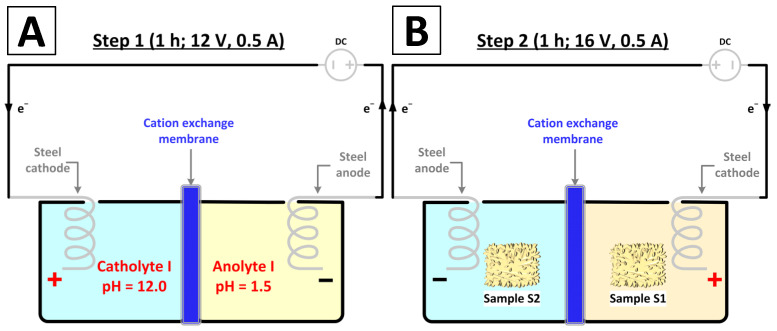
Schematic illustration of the experimental setup for step 1 (**A**) and step 2 (**B**) of the iron compound deposition process.

**Figure 4 biomimetics-09-00387-f004:**
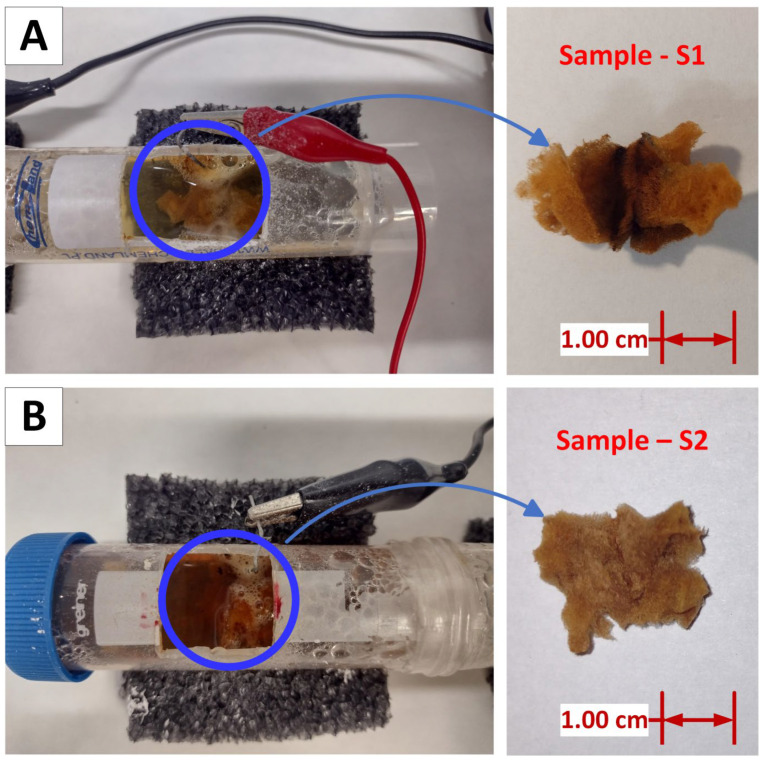
Spongin scaffolds after the second step (1 h; 16 V, 0.5 A) of electro-supported deposition of iron compounds: sample S1 (cathode chamber) (**A**) and sample S2 (anode chamber) (**B**).

**Figure 5 biomimetics-09-00387-f005:**
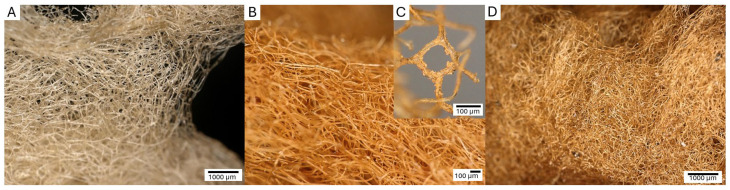
Comparative microscopic analysis of spongin fibers. (**A**) Untreated spongin fibers show a natural complex lattice. Treated samples S1 (**B**) and S2 (**D**) show a uniform rust coloration, indicating iron oxide deposition. (**C**) A magnified view of the treated S1 fiber reveals irregular pale-yellow microdeposits that are tightly bound to the surface of the spongin.

**Figure 6 biomimetics-09-00387-f006:**
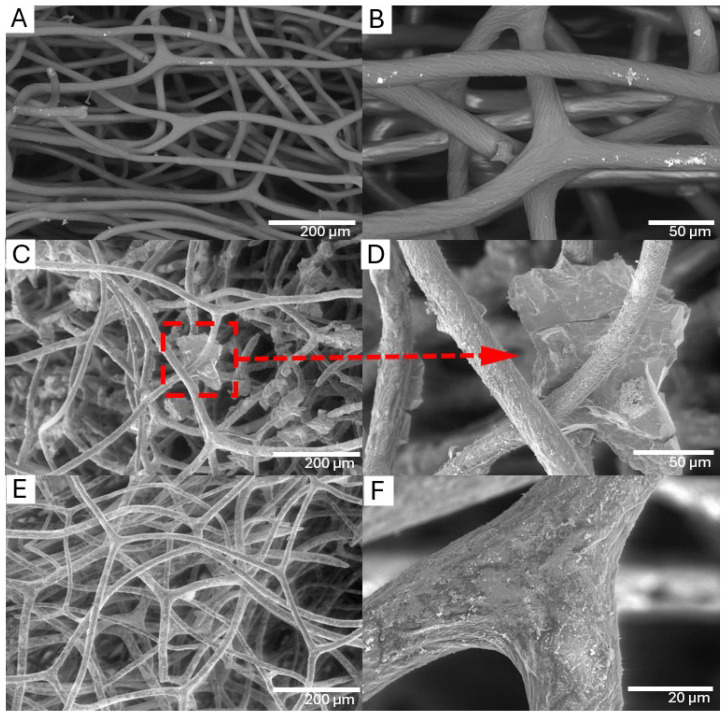
SEM images of pure spongin and spongin after electro-supported deposition of iron compounds. Untreated spongin fibers (**A**,**B**). Sample S1 with localized iron oxide crystallization (**C**,**D**). Sample S2 with homogenous iron oxide coating (**E**,**F**). Scale bars: (**A**,**C**,**E**)—200 µm; (**B**,**D**)—50 µm; (**F**)—20 µm.

**Figure 7 biomimetics-09-00387-f007:**
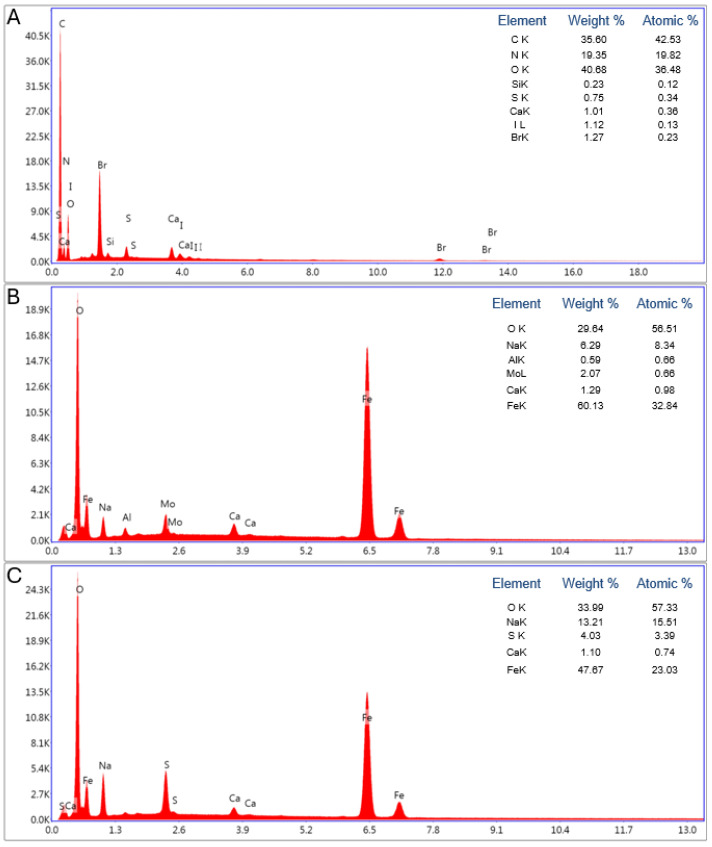
Energy-dispersive X-ray (EDX) spectral analysis of spongin samples: untreated spongin (**A**), sample S1 (**B**), and sample S2 (**C**).

**Figure 8 biomimetics-09-00387-f008:**
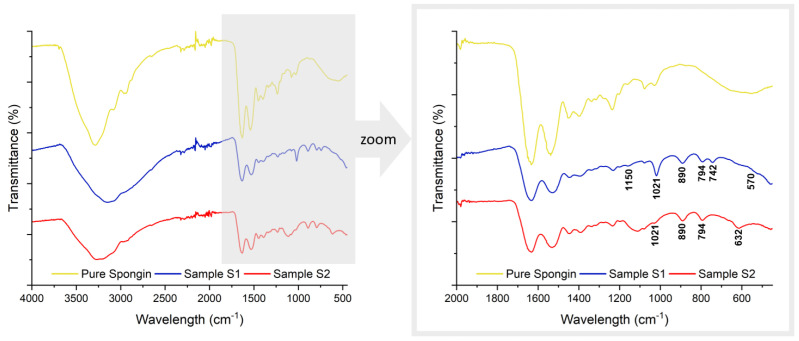
FTIR spectral comparison. The full-band FTIR spectra for pure spongin, sample S1, and sample S2, with a magnified view of the 2000 to 450 cm^−1^ region, detailing the absorption bands associated with iron oxides in S1 and S2.

**Figure 9 biomimetics-09-00387-f009:**
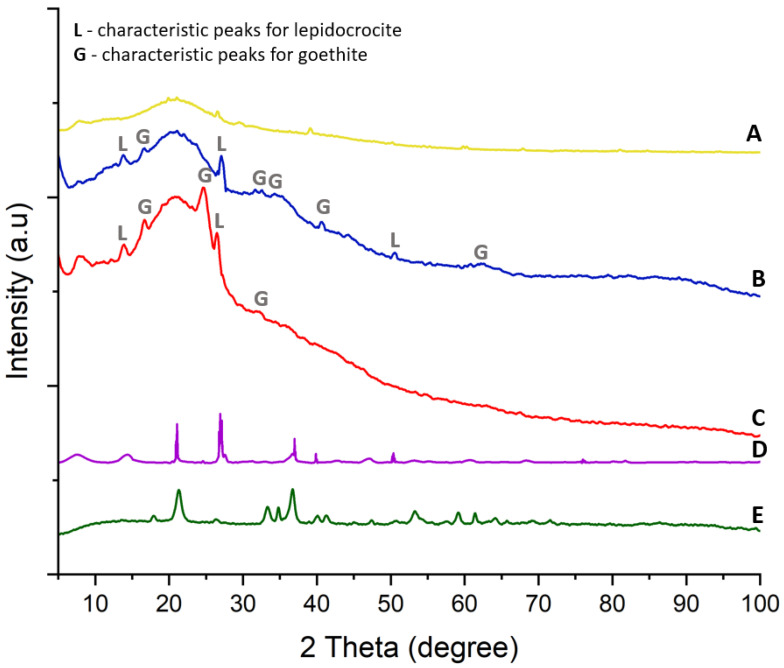
XRD analysis of spongin samples and standards. Pure *H. communis* spongin (**A**). Sample S1 (**B**). Sample S2 (**C**). Lepidocrocite standard (**D**). Goethite standard (**E**).

**Table 1 biomimetics-09-00387-t001:** Wavenumbers and assignments of bands in the studied spectra.

Pure Spongin	Sample S1	Sample S2	Vibrational Assignment
3410	3408	3410	–OH stretching
3300	3300	3303	–NH stretching
-	3143	3140	–OH stretching
2931	2929	2933	–CH2, –CH3 stretching
1633	1633	1633	C=O stretching
1536	1536	1537	–NH deformational
1244	1245	1244	C–N stretching
-	1150	-	–OH deformational
1030	-	-	C–O stretching
-	1021	-	Fe–OH
-	890	890	–OH bending
-	794	794	–OH bending
-	742	-	–OH deformational
-	635	632	Fe–O stretching
-	570	-	Fe–O stretching
472	460	462	N–H stretching

## Data Availability

The data presented in this study are available on request from the corresponding authors.

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
