# Peer review of "3D Spongin Scaffolds as Templates for Electro-Assisted Deposition of Selected Iron Oxides"

_biomimetics, 2024, doi:10.3390/biomimetics9070387_

Round 1

Reviewer 1 Report

Comments and Suggestions for Authors

This is an interesting study, which deals with electro-deposition and formation of iron-oxide on the surface of spongin, a scleroprotein scaffold of natural sponges. This paper is recommended to be accepted for publication after clarifying certain insufficient details in the submitted manuscript as commented below.

The EDX spectra of the samples, including the “pure” sponging, with the detailed spectra analyses have to be presented. Otherwise, the statements and data in lines 219-226 are not supported at all in this manuscript.

In lines 261-262, the authors write that “The spongin baseline (Figure 8A) shows a largely featureless 261 curve with a broad hump and no sharp peaks”. However, the curve in Figure 8A shows very well detectable small peaks indicating the presence of iron oxide in the “pure” sample. Therefore, one might conclude that the formation of crystalline iron oxide in the sponging is due to the presence of the small amount of iron oxide crystals in this material as crystal seeds. Did the authors carried out control experiments by adding aqueous iron(III) sulfate(VI) to sponging kept in water at pH 1.5 and 12? If so, did they determine the amount and type of iron oxide crystals formed in the sponging?

As to the FTIR spectra, it should be noted that the peaks are very weak, and insufficient for solid conclusions. This also indicates the importance to present the EDX spectra for the readers.

Author Response

Reviewer #1

Comment: This is an interesting study, which deals with electro-deposition and formation of iron-oxide on the surface of spongin, a scleroprotein scaffold of natural sponges. This paper is recommended to be accepted for publication after clarifying certain insufficient details in the submitted manuscript as commented below.

Answer: Thank you very much for your positive feedback and constructive comments on our study. We have made every effort to address all the concerns raised and have revised the manuscript accordingly.

Query: The EDX spectra of the samples, including the “pure” spongin, with the detailed spectra analyses have to be presented. Otherwise, the statements and data in lines 219-226 are not supported at all in this manuscript.

Answer: Thank you for your valuable suggestion. In response to your comment, we have included the EDX spectra of the pure spongin and the two treated samples (S1 and S2) – Figure 7. This addition has strengthened the analysis by allowing a direct comparison of the spectra, thereby providing better support for the statements and data presented in the manuscript.

Query: In lines 261-262, the authors write that “The spongin baseline (Figure 8A) shows a largely featureless 261 curve with a broad hump and no sharp peaks”. However, the curve in Figure 8A shows very well detectable small peaks indicating the presence of iron oxide in the “pure” sample. Therefore, one might conclude that the formation of crystalline iron oxide in the sponging is due to the presence of the small amount of iron oxide crystals in this material as crystal seeds.

Respond: Thank you for your insightful comment. We have indeed overlooked the presence of small peaks in the curve shown in Figure 8A, which indicate the presence of iron oxide in the "pure" sample. In natural environments, marine sponges exhibit the presence of lepidocrocite with small amounts of goethite. We have now updated the text to acknowledge and highlight these peaks.

We suggest that the formation mechanism of lepidocrocite/goethite in our samples mirrors the natural processes observed in marine sponges. This mechanism, which we proposed in our previous work [1], involves the interaction of functional groups of amino acid sequences in spongin fibers with iron ions. Cysteine, in particular, facilitates the transformation of iron phases due to its ability to form disulfide bonds. Additionally, the environment further promotes  the development of stable iron mineral phases like lepidocrocite or goethite.

  1. Kubiak et al. (2023) Spongin as a Unique 3D Template for the Development of Functional Iron-Based Composites Using Biomimetic Approach In Vitro. Drugs, 21, 460. DOI: 10.3390/md21090460

Query: Did the authors carried out control experiments by adding aqueous iron(III) sulfate(VI) to spongin kept in water at pH 1.5 and 12? If so, did they determine the amount and type of iron oxide crystals formed in the spongin?

Respond: In this study, we did not carry out control experiments adding aqueous iron(III) sulphate to sponges stored in water at pH 1.5 and 12. However, your proposal is interesting for the future research on this topic. In our previous study [1], we focused on reproducing the natural process of lepidocrocite formation under laboratory conditions, using seawater (pH 8.2) and iron powder. It took 30 days under these conditions to achieve a similar degree of iron oxide coverage on spongin fibres as described in the current article. In designing the current experiment, our aim was to speed up the process while maintaining a similar result. Given the extended time frame, we decided not to carry out additional experiments without using electrolysis under extreme pH conditions.

We appreciate the understanding and will consider such control experiments in future studies to further investigate the impact of pH changes.

  1. Kubiak et al. (2023) Spongin as a Unique 3D Template for the Development of Functional Iron-Based Composites Using Biomimetic Approach In Vitro. Drugs, 21, 460. DOI: 10.3390/md21090460

Query: As to the FTIR spectra, it should be noted that the peaks are very weak, and insufficient for solid conclusions. This also indicates the importance to present the EDX spectra for the readers.

Respond: Thank you for your critical remark regarding the FTIR spectra.

While we understand that some peaks in the FTIR spectra might appear weak, we believe they still provide significant information about the molecular interactions and functional groups present in our samples. The FTIR analysis in our study has revealed distinct absorption bands corresponding to the functional groups of spongin and iron oxides.

The spectra for samples S1 and S2 show notable new bands that are characteristic for iron oxyhydroxides and iron oxides. These bands, particularly those associated with goethite and lepidocrocite, indicate the successful integration of corresponding iron oxides into the spongin structure. The intensity and clarity of these bands, especially in sample S1, suggest a significant presence of well-crystallized iron oxides, which is crucial for our conclusions.

To provide a comprehensive understanding of the sample composition, we have also included Energy Dispersive X-ray (EDX) spectra prior to FTIR data in the manuscript. The EDX spectra complement the FTIR data by offering detailed elemental analysis, thereby strengthening our findings. We appreciate your valuable feedback and trust that the combination of FTIR and EDX data offers a robust and thorough characterization of our samples.

Reviewer 2 Report

Comments and Suggestions for Authors

Manuscript title: 3D spongin scaffolds as templates for electro-assisted deposition of selected iron oxides

Summary:

This work provides an electro-assisted deposition on the 3D microporous spongin scaffold to fabricate metal-polymer composite, which is motivated by the interaction between iron ion and spongin. The composite has been fully characterized by SEM, FTIR, and X-ray diffraction. The paper is routine, scientifically solid, and clearly written. Therefore, the reviewer suggests the acceptance of the paper for publication after addressing some minor comments.

Comments:

1. Page 2 paragraph 3-5: the reviewer suggests to combine these three paragraphs to concisely introduce the application of electrodeposition and its use in this work.

2. Page 6 line 176: The tile of result section should be result not method terminology. For example, here it could be "Microstructure of spongin scaffold" This comment applies to the following sections.

3. Page 10 paragraph 2: This paragraph seems to be an important motivation of this work, which is to enhance the efficiency of synthesize the iron-spongin composite. Therefore, it should be introduced at the beginning.

Author Response

Reviewer #2

Summary:

This work provides an electro-assisted deposition on the 3D microporous spongin scaffold to fabricate metal-polymer composite, which is motivated by the interaction between iron ion and spongin. The composite has been fully characterized by SEM, FTIR, and X-ray diffraction. The paper is routine, scientifically solid, and clearly written. Therefore, the reviewer suggests the acceptance of the paper for publication after addressing some minor comments.

Comments:

Query 1. Page 2 paragraph 3-5: the reviewer suggests to combine these three paragraphs to concisely introduce the application of electrodeposition and its use in this work.

Respond: Thank you for your suggestion. Changes have been made into the manuscript.

Query 2. Page 6 line 176: The tile of result section should be result not method terminology. For example, here it could be "Microstructure of spongin scaffold" This comment applies to the following sections.

Respond: Thank you for your insightful comment. Titles of referred sections have been changed.

Query 3. Page 10 paragraph 2: This paragraph seems to be an important motivation of this work, which is to enhance the efficiency of synthesize the iron-spongin composite. Therefore, it should be introduced at the beginning.

Query: Thank you for this important remark.  The manuscript has been modified. 

Reviewer 3 Report

Comments and Suggestions for Authors

Dear Authors,

Regarding your article entitled " 3D sponging scaffolds ...", submitted to  Biomimetic Journal, I would like to inform you that I found the article adia interesting and you studied it carefully and successfully.

You need to address the applications of the final products in the abstract section.

Accordingly, I accepted the manuscript after minor corrections

Author Response

Reviewer #3

Dear Authors,

Regarding your article entitled " 3D sponging scaffolds ...", submitted to  Biomimetic Journal, I would like to inform you that I found the article idia interesting and you studied it carefully and successfully.

Query: You need to address the applications of the final products in the abstract section.

Respond: Thank you for your insightful comment. The possible final applications of iron oxide- spongin composites have been added to the abstract section.

Accordingly, I accepted the manuscript after minor corrections.